# An Adaptive Filtering Method for Cooperative Localization in Leader–Follower AUVs

**DOI:** 10.3390/s22135016

**Published:** 2022-07-02

**Authors:** Lin Zhao, Hong-Yi Dai, Lin Lang, Ming Zhang

**Affiliations:** 1College of Intelligence Science and Technology, National University of Defense Technology, Changsha 410073, China; zhaolin689@163.com (L.Z.); langlin_8502@nudt.edu.cn (L.L.); 2College of Science, National University of Defense Technology, Changsha 410073, China; daihongyi1@163.com

**Keywords:** multi-AUV cooperative localization, measurement anomaly, adaptive filter, extended Kalman filter

## Abstract

In the complex and variable marine environment, the navigation and localization of autonomous underwater vehicles (AUVs) are very important and challenging. When the conventional Kalman filter (KF) is applied to the cooperative localization of leader–follower AUVs, the outliers in the sensor observations will have a substantial adverse effect on the localization accuracy of the AUVs. Meanwhile, inaccurate noise covariance matrices may result in significant estimation errors. In this paper, we proposed an improved Sage–Husa adaptive extended Kalman filter (improved SHAEKF) for the cooperative localization of multi-AUVs. Firstly, the measurement anomalies were evaluated by calculating the Chi-square test statistics based on the innovation. The detection threshold was determined according to the confidence level of the Chi-square test, and the Chi-square test statistics exceeding the threshold were regarded as measurement abnormalities. When measurement anomalies occurred, the Sage–Husa adaptive extended Kalman filter algorithm was improved by suboptimal maximum a posterior estimation using weighted exponential fading memory, and the measurement noise covariance matrix was adjusted online. The numerical simulation of leader–follower multi-AUV cooperative localization verified the effectiveness of the improved SHAEKF and demonstrated that the average root mean square and the average standard deviation of the localization errors based on the improved SHAEKF were significantly reduced in the case of the presence of measurement abnormalities.

## 1. Introduction

The Global Navigation Satellite System (GNSS) is often unavailable in deep water environments, and inertial/dead reckoning (DR) navigation technology is usually considered. The Doppler velocity log (DVL) can measure the velocity of autonomous underwater vehicle (AUVs) and it can carry out autonomous navigation and localization without installing additional sensors and external reference information when the DVL is combined with the Inertial Navigation System (INS). In a multi-AUV formation, there will be localization error growth over time for the AUVs, which are only equipped with inertial/DR equipment. We can improve the localization accuracy of the whole system by improving the localization accuracy of each AUV, but there is a great sacrifice in cost. Therefore, researchers have proposed leader–follower multi-AUV cooperative localization methods [1,2,3]. AUVs transmit information through sensors, such as underwater acoustic communication equipment. When conducting cooperative localization, various state estimation technologies need to be introduced to estimate the position of the AUV. A Kalman filter (KF) is the best Bayesian estimator for linear systems with Gaussian uncertainty [4]. However, the model of a leader–follower multi-AUV cooperative localization system is often nonlinear. Therefore, an extended Kalman filter (EKF) or unscented Kalman filter (UKF) is usually used for state estimation. In the framework of Kalman filters, these methods require that the statistical characteristics of process noise and measurement noise are known and fixed during cooperative localization. However, in practical applications, due to the low precision of low-cost inertial/DR equipment carried on an AUV, these methods are easily affected by many uncertain factors such as external environment interference, carrier maneuver changes, internal instrument failure, and so on. At present, in order to overcome the uncertainty of the filter in AUV navigation and localization, many researchers have applied adaptive interactive multi-models, fuzzy logic, and other methods [5,6,7,8,9,10]. In Refs. [5,6], the adaptive filtering algorithm was applied to an AUV. When the system noise matrix remained positive or semi positive, the latest measurement data were used to adaptively estimate the changing process noise or measurement noise covariance matrix. Refs. [7,8] introduced interactive multi-model methods in order to solve the problem of state estimation when the statistical characteristics of measurement noise were unknown or easy to change under the complex working environment of an AUV, and the model probability obtained in the process of state estimation was used for decision-making, so as to obtain the expected model closer to the real mode of the system for state estimation. In refs. [9,10], researchers used fuzzy logic technology to adapt to sensor noise changes or communication data loss, enhanced the fault detection and signal recovery algorithm of the navigation system reliability, adjusted the initial noise statistical assumption of the EKF, and maintained the stability and performance of the filter. In the aforementioned research, the improved filter algorithm was mostly used for navigation and localization of a single AUV. This study was concentrated on the improved filter algorithm for the navigation and localization of leader–follower AUVs. In this paper, an improved Sage–Husa adaptive extended Kalman filter (improved SHAEKF) was proposed to improve the adaptability of the filter and the accuracy of cooperative localization.

## 2. Multi-AUV Cooperative Localization Method Based on an EKF

### 2.1. State Model

The navigation motion of AUV formation in a GNSS-denied environment is a complex three-dimensional process. If the formation adopts the leader–follower multi-AUV structure, the system includes multiple leader AUVs and follower AUVs; a leader AUV and a follower AUV exchange and share information through underwater acoustic communication equipment [11]. After the follower AUV obtains the cooperative localization information, the cooperative localization filtering algorithm is used to correct position information. In order to facilitate the study of cooperative localization algorithms with measurement anomalies, the state variables of a leader AUV (the *l*th leader AUV) and a follower AUV (the *j*th follower AUV) are analyzed by a simplified level equation under reasonable assumptions, and the non-rotating east, north, and up (ENU) geographic reference system ${O_{g}}^{\prime }-{X_{g}}^{\prime }{Y_{g}}^{\prime }{Z_{g}}^{\prime }$ is used to analyze the state variables, as shown in Figure 1a.

Assuming that the depth of an AUV can be measured directly and accurately by depth gauge, the discrete-time nonlinear kinematics equations of an AUV are as follows:(1)$$\left\{\begin{array}{ccc} x_{k+1} & = & x_{k}+\Delta TV_{k}\sin\psi_{k}\\ y_{k+1} & = & y_{k}+\Delta TV_{k}\cos\psi_{k}\\ \psi_{k+1} & = & {\overset{⌢}{\;\psi }}_{k+1}\\ V_{k+1} & = & {\overset{⌢}{\;V}}_{k+1} \end{array}\right.$$
where *x*, *y* respectively represent the coordinates of the longitude and latitude position of the AUV converted into the geographical coordinate system, and ψ, *V*, ΔT respectively represent the heading angle, forward synthetic motion velocity, and sampling period. Moreover, in this paper, the quantity with superscript “ ⌢ ” represents the observed value of the correlation quantity, and the quantity with superscript “^” represents the predicted value of the correlation quantity.

The state vector X and the control input vector u of the *j*th follower AUV at time *k* are defined as follows:(2)$$\left\{\begin{array}{c} X_{k}^{j}={\left[\begin{array}{cccc} x_{k}^{j} & y_{k}^{j} & \psi_{k}^{j} & V_{k}^{j} \end{array}\right]}^{T}\\ u_{k}^{j}={\left[\begin{array}{cc} \psi_{k}^{j} & V_{k}^{j} \end{array}\right]}^{T} \end{array}\right.$$
where the right superscript “*T*” represents the transpose mode of the vector matrix.

The observed values of the control input are as follows:(3)$$\left\{\begin{array}{c} {\overset{⌢}{\;\psi }}_{k}^{j}=\psi_{k}^{j}+w_{k,\psi }^{j}\\ {\overset{⌢}{\;V}}_{k}^{j}=V_{k}^{j}+w_{k,V}^{j} \end{array}\right.$$
where $w_{k,\psi }^{j}$, $w_{k,V}^{j}$ represent the observation errors of the heading angle and the forward synthetic motion velocity, respectively. It is assuming that $w_{k,\psi }^{j}$ and $w_{k,V}^{j}$ are Gaussian-white-noise independent of each other and the variances are ${\left(\sigma_{k,\psi }^{j}\right)}^{2}$ and ${\left(\sigma_{k,V}^{j}\right)}^{2}$, respectively.

When $w_{k}^{j}={\left[\begin{array}{cc} w_{k,\psi }^{j} & w_{k,V}^{j} \end{array}\right]}^{T}$ is defined as process noise, $E\left(w_{k}^{j}{\left(w{}_{k}^{j}\right)}^{T}\right)=Q_{k}^{j}\cdot \delta_{pq}$, where $\delta_{pq}$ represents the Dirac delta function, $\delta_{pq}=\left\{\begin{array}{cc} 1 & if_{}\;p=q\\ 0 & otherwise \end{array}\right.$. The process noise covariance matrix is $Q_{k}^{j}$ and the state model equation of the *j*th follower AUV can be expressed as follows:(4)$$X_{k+1}^{j}=f\left(X_{k}^{j},u_{k}^{j},w_{k}^{j}\right)$$
where f(·) is a nonlinear function.

### 2.2. Measurement Model

In this study, the cooperative localization information mainly included the relative distance and relative direction angle between AUVs and the position coordinates of the leader AUV. The follower AUV established the measurement equation by obtaining the cooperative localization information shared by the leader AUV. Figure 1b depicts the geometric relationship between the relative distance $r^{lj}$ and the relative direction angle $\alpha^{lj}$ of the *l*th leader AUV and the *j*th follower AUV. It was assumed that the forward movement direction of the follower AUV was consistent with the positive direction of the ${Y_{b}}^{\prime }$-axis of its own carrier coordinate system. From the positive direction line of the ${Y_{b}}^{\prime }$-axis, the relative direction angle was positive when moving clockwise to the relative direction angle line, otherwise, it was negative. Assuming that the position coordinates of the *l*th leader AUV at time *k* is ${\left[\begin{array}{cc} {x^{L}}_{k}^{l} & {y^{L}}_{k}^{l} \end{array}\right]}^{T}$, the geometric relation equations at time *k* are as follows:(5)$$\left\{\begin{array}{c} r_{k}^{lj}=\sqrt{{\left(r_{k,x}^{lj}\right)}^{2}+{\left(r_{k,y}^{lj}\right)}^{2}}\\ \alpha_{k}^{lj}=\frac{\pi }{2}-\psi_{k}^{j}-\tan^{-1}\left(\frac{r_{k,y}^{lj}}{r_{k,x}^{lj}}\right) \end{array}\right.$$
where $r_{k,x}^{lj}={x^{L}}_{k}^{l}-x_{k}^{j}$, $r_{k,y}^{lj}={y^{L}}_{k}^{l}-y_{k}^{j}$ respectively represent the decomposition of the true relative distance $r^{lj}$ of the AUV in the direction of the ${X_{g}}^{\prime }$-axis and the ${Y_{g}}^{\prime }$-axis of the geographical coordinate system. The observed values of the relative distance and relative direction angle between the *j*th follower AUV and the *l*th leader AUV are as follows:(6)$$\left\{\begin{array}{c} {\overset{⌢}{\;r}}_{k}^{lj}=r_{k}^{lj}+v_{k,r}^{lj}\\ {\overset{⌢}{\;\alpha }}_{k}^{lj}=\alpha_{k}^{lj}+v_{k,\alpha }^{lj} \end{array}\right.$$
where $v_{k,r}^{lj}$ and $v_{k,\alpha }^{lj}$ respectively represent the observation errors of the relative distance and relative direction angle between the *j*th follower AUV and the *l*th leader AUV. It is assumed that $v_{k,r}^{lj}$ and $v_{k,\alpha }^{lj}$ are Gaussian-white-noise independent of each other, and the variances are ${\left(\sigma_{k,r}^{lj}\right)}^{2}$ and ${\left(\sigma_{k,\alpha }^{lj}\right)}^{2}$, respectively.

The observed value ${\overset{⌢}{r}}_{k}^{lj}$ of the relative distance is decomposed along the ${X_{b}^{j}}^{\prime }$-axis and the ${X_{b}^{j}}^{\prime }$-axis of the carrier coordinate system of the *j*th follower AUV to obtain the following [12]:(7)$$\left\{\begin{array}{c} {\overset{⌢}{r}}_{k,xb}^{lj}={\overset{⌢}{r}}_{k}^{lj}\sin{\overset{⌢}{\alpha }}_{k}^{lj}\\ {\overset{⌢}{r}}_{k,yb}^{lj}={\overset{⌢}{r}}_{k}^{lj}\cos{\overset{⌢}{\alpha }}_{k}^{lj} \end{array}\right.$$

If (Equation 6) is substituted into (Equation 7) and we make $\cos v_{k,\alpha }^{lj}=1$, $\sin v_{k,\alpha }^{lj}=v_{k,\alpha }^{lj}$, $v_{k,r}^{lj}\cdot v_{k,\alpha }^{lj}=0$, the following formulas are obtained by simplification:(8)$$\begin{array}{c} \left\{\begin{array}{c} {\overset{⌢}{r}}_{k,xb}^{lj}=r_{k,xb}^{lj}+v_{k,r}^{lj}\sin\left(\alpha_{k}^{lj}\right)+r_{k}^{lj}v_{k,\alpha }^{lj}\cos\left(\alpha_{k}^{lj}\right)=r_{k,xb}^{lj}+v_{k,xb}^{lj}\\ {\overset{⌢}{r}}_{k,yb}^{lj}=r_{k,yb}^{lj}+v_{k,r}^{lj}\cos\left(\alpha_{k}^{lj}\right)-r_{k}^{lj}v_{k,\alpha }^{lj}\sin\left(\alpha_{k}^{lj}\right)=r_{k,yb}^{lj}+v_{k,yb}^{lj} \end{array}\right. \end{array}$$

${\overset{⌢}{Z}}_{k+1}^{lj}={\left[\begin{array}{cc} {\overset{⌢}{r}}_{k+1,xb}^{lj} & {\overset{⌢}{r}}_{k+1,yb}^{lj} \end{array}\right]}^{T}$ is defined as the observation value of the measurement model based on the observation of relative distance and relative direction angle, and it is assumed that $Z_{k+1}^{lj}\approx {\overset{⌢}{Z}}_{k+1}^{lj}$. $v_{k}^{lj}={\left[\begin{array}{cc} v_{k,xb}^{lj} & v_{k,yb}^{lj} \end{array}\right]}^{T}$ is the observation noise of the measurement model based on the observation of relative distance and relative direction angle, the covariance matrix is $R_{k,Z^{lj}}^{lj}=E\left(v_{k}^{j}{\left(v{}_{k}^{j}\right)}^{T}\right)$. The nonlinear measurement model equation of the *j*th follower AUV is established as follows:(9)$$Z_{k+1}^{lj}=h\left(X_{k+1}^{j}\right)+v_{k+1}^{lj}=\left[\begin{array}{c} r_{k+1,xb}^{lj}\\ r_{k+1,yb}^{lj} \end{array}\right]+v_{k+1}^{j}$$
where h(·) is a nonlinear function.

### 2.3. Cooperative Localization Algorithm Based on an EKF

From (Equation 4) and (Equation 9), we obtain the discrete-time nonlinear state equation and measurement equation of the *j*th follower AUV as follows:(10)$$\left\{\begin{array}{ccc} X_{k+1}^{j} & = & f(X_{k}^{j},u_{k}^{j},w_{k}^{j})\\ Z_{k+1}^{lj} & = & h\left(X_{k+1}^{j}\right)+v_{k+1}^{lj} \end{array}\right.$$

The EKF algorithm is used for state estimation. The steps in a single filtering cycle are as follows.

Step 1 (state prior estimation):(11)$${\hat{X}}_{k+1|k}^{j}=f\left({\hat{X}}_{k}^{j},{\hat{u}}_{k}^{j},0\right)$$

Step 2 (the innovation sequence update):(12)$$s_{k+1}^{lj}=Z_{k+1}^{lj}-h\left({\hat{X}}_{k+1|k}^{j}\right)$$

Step 3 (error covariance updating of state a priori estimation):(13)$$P_{k+1|k}^{j}=\Phi_{k}^{j}P_{k}^{j}{\left(\Phi_{k}^{j}\right)}^{T}+\Gamma_{k}^{j}Q_{k}^{j}{\left(\Gamma_{k}^{j}\right)}^{T}$$
where $\Phi_{k}^{j}$ is the Jacobian matrix of *f* with respect to $X_{k}^{j}$, $\Gamma_{k}^{j}$ is the Jacobian matrix of *f* with respect to $u_{k}^{j}$.

Step 4 (filter gain update):(14)$$K_{k+1}^{j}=P_{k+1|k}^{j}{\left(H_{k+1}^{j}\right)}^{T}{\left(H_{k+1}^{j}P_{k+1|k}^{j}{\left(H_{k+1}^{j}\right)}^{T}+R_{k+1}^{lj}\right)}^{-1}$$
where $H_{k+1}^{j}$ is the Jacobian matrix of *h* with respect to ${\hat{X}}_{k+1|k}^{j}$.

Step 5 (state a posteriori estimation update):(15)$$X_{k+1}^{j}={\hat{X}}_{k+1|k}^{j}+K_{k+1}^{j}s_{k+1}^{lj}$$

Step 6 (error covariance of state a posteriori estimation):(16)$$P_{k+1}^{j}=\left(I-K_{k+1}^{j}H_{k+1}^{j}\right)P_{k+1|k}^{j}$$
where I is the identity matrix.

## 3. Adaptive Cooperative Localization Algorithm

Ref. [13] analyzes the position error covariance of a group of agents equipped with an inertial measurement unit (IMU) and relative distance and azimuth sensors. Researchers regard all the agents in the group as a unified system. Agents exchange their information, including measured and estimated location data. Therefore, the exchange of each external perceptual measurement results in an improvement in the overall position error estimation. The analysis showed that the system position error covariance increased with the cube of time. Accordingly, as the number of agents increased, the growth rate of covariance decreased. However, due to the different construction methods of the system state equation and the measurement equation, the conclusions obtained from the analysis were also different. For example, in the agent equipped with an IMU, the covariance growth of position error did not depend on the path. However, the position error covariance of an agent equipped with an encoder depended on the path and changed with the path direction.

Refs. [12,14] studied the transfer equation of the overall localization uncertainty with respect to the relative position measurement error. By solving the algebraic Riccati equation of the evolution of the system localization errors with time, the analytical expression of the upper bound of the expected positioning uncertainty was determined. The analysis of ref. [12] shows that the upper bound of the variance of the localization errors in the steady state depended on the measurement accuracy of the AUV’s speed, heading angle, and relative position, and was independent of the initial filter variance of the system.

All the above studies were under the steady state of the system. We know that the noise error of an AUV sensor is related to the process noise error and measurement noise error in the cooperative localization algorithm based on an EKF, but there are differences between process noise error and measurement noise error. Through error uncertainty propagation analysis [12,13,14], it was found that the covariance matrix of system process noise and measurement noise were time-varying.The process noise covariance matrix $Q_{k}^{j}$ is related to the observation value of heading angle, which is mainly determined by the internal mechanism of a system and is relatively stable. The measurement noise covariance matrix $R_{k+1}^{lj}$ is not only related to the error covariance matrix $R_{k,Z^{lj}}^{lj}$ of the measurement model, but also to the observation value of heading angle and the noise of the external information, which is easy to change, and there is a certain unpredictability. In the model of the leader–follower multi-AUV cooperative localization system studied in this paper, we introduced and improved the SHAEKF to help solve the adverse impact of the follower AUV measurement anomalies on localization. A measurement anomaly might occur in the prior characteristics of measurement noise based on relative distance and relative heading angle, or when the leader AUV transmits position data to the follower AUV. This problem has received less attention in previous research on cooperative localization algorithms.

### 3.1. Sage–Husa Adaptive Kalman Filter Algorithm

The Sage–Husa adaptive filtering algorithm is an improvement based on the classical Kalman filtering algorithm. The Sage–Husa filter adaptively estimates system process noise and measurement noise online, including the mean vector ${\hat{q}}_{k}$, the covariance matrix ${\hat{Q}}_{k}$ of the process noise, the mean vector ${\hat{r}}_{k}$, and the covariance matrix ${\hat{R}}_{k}$ of the measurement noise. The estimators of the Sage–Husa adaptive filtering algorithm can be expressed as follows [6,15]:(17)$$\left\{\begin{array}{cc} {\hat{q}}_{k+1} & =\left(1-\frac{1}{k+1}\right){\hat{q}}_{k}+\frac{1}{k+1}\left(X_{k+1}-{\hat{X}}_{k+1|k}\right)\\ \Gamma_{k+1}{\hat{Q}}_{k+1}{\left(\Gamma_{k+1}\right)}^{T} & =\left(1-\frac{1}{k+1}\right)\Gamma_{k}{\hat{Q}}_{k}{\left(\Gamma_{k}\right)}^{T}+\frac{1}{k+1}(K_{k+1}s_{k+1}{\left(s_{k+1}\right)}^{T}{\left(K_{k+1}\right)}^{T}\\ \; & +P_{k+1|k}-\Phi_{k+1}P_{k+1|k}{\left(\Phi_{k+1}\right)}^{T})\\ {\hat{r}}_{k+1} & =\left(1-\frac{1}{k+1}\right){\hat{r}}_{k}+\frac{1}{k+1}\left(Z_{k+1}-h\left(X_{k+1}\right)\right)\\ {\hat{R}}_{k+1} & =\left(1-\frac{1}{k+1}\right){\hat{R}}_{k}+\frac{1}{k+1}\left(s_{k+1}{\left(s_{k+1}\right)}^{T}-H_{k+1}P_{k+1|k}{\left(H_{k+1}\right)}^{T}\right) \end{array}\right.$$

Since the innovation vector affects the calculation of ${\hat{Q}}_{k}$ and ${\hat{R}}_{k}$ at the same time, this easily led to filter divergence. Therefore, the noise estimator could not estimate the statistical properties of the process noise and measurement noise at the same time. Moreover, there was a negative sign in the above formula and the positive definite of ${\hat{Q}}_{k}$ and ${\hat{R}}_{k}$ could not be fully guaranteed.

We focused on finding a method to detect an anomaly and re-estimate the statistical characteristics of the measurement noise when the measurement noise was abnormal, which is discussed next in this paper.

### 3.2. Measurement Anomaly Detection

This innovation can reflect the relationship between the observed value and the estimated value of a measurement model, so it is often used to measure the filtering performance of a Kalman filter [16,17]. According to Kalman filter theory, if the discrete-time nonlinear state space model of process and measurement noise is established under the assumption of Gaussian distribution and the measurement is not subject to abnormal interference, the innovation sequence obeys the standard Gaussian distribution. The expectation and covariance of the innovation have the following characteristics:(18)$$\left\{\begin{array}{c} E[s_{k+1}^{lj}]=0\\ E\left[s_{k+1}^{lj}{\left(s_{i+1}^{lj}\right)}^{T}\right]=\left\{\begin{array}{cc} H_{k+1}^{j}P_{k+1|k}^{j}{\left(H_{k+1}^{j}\right)}^{T}+R_{k+1}^{lj}=P_{s_{k+1}^{lj}} & k=i\\ 0 & k\neq i \end{array}\right. \end{array}\right.$$

The probability density function of the innovation can be written in the following form:(19)$$f_{s^{lj}}\left(s_{k+1}^{lj}\right)=\frac{1}{\sqrt{{\left(2\pi \right)}^{m}\left|P_{s_{k+1}^{lj}}\right|}}exp\left(-\frac{1}{2}{\left(s_{i+1}^{lj}\right)}^{T}{\left(P_{s_{k+1}^{lj}}\right)}^{-1}s_{k+1}^{lj}\right)$$
where *m* is the degree of freedom, and $\left|P_{s_{k+1}^{lj}}\right|$ is the determinant of $P_{s_{k+1}^{lj}}$.

If there are some observed outliers in the measurement model, or the measurement noise is affected by other noises and no longer conforms to the Gaussian distribution, (Equation 18) and (Equation 19) will no longer be tenable. It can be considered that there are some violations of assumptions or some modeling errors. Specifically, hypothesis testing can be performed to detect measurement anomalies. The purpose of hypothesis testing is to check whether the actual measurement is compatible with the hypothetical model or, in other words, the zero hypothesis. According to the orthogonality principle of the innovation in Kalman filtering, the square of Mahalanobis distance based on innovation obeys the Chi-square distribution. We can took (Equation 19) as the relevant zero distribution that did hold under the hypothetical model, and then constructed the test statistics based on the following innovation:(20)$$\xi_{k+1}={\left(s_{k+1}^{lj}\right)}^{T}{\left(P_{s_{k+1}^{lj}}\right)}^{-1}s_{k+1}^{lj}$$

$\xi_{k+1}$ follows the Chi-square distribution with degree of freedom *m* if the assumption holds. The significance level is set as γ, which indicates the probability threshold that the null hypothesis below this threshold will be rejected. The critical value of the corresponding Chi-square test is $\chi_{\gamma }^{2}\left(m\right)$. When the innovation sequence is calculated with the actual observation value, the actual judgment index ${\overset{⌢}{\;\xi }}_{k+1}$ is obtained. When ${\overset{⌢}{\;\xi }}_{k+1}>\chi_{\gamma }^{2}\left(m\right)$ rejects the original hypothesis, it is considered that the measurement is abnormal. For example, from the Chi-square distribution table, we obtained that the probability of $\left\{\chi_{\gamma }^{2}\left(m=3\right)>11.345\right\}$ was only 1%, i.e., γ=0.01. The problem of measurement anomaly detection is expressed as:(21)$$H_{0}:{\overset{⌢}{\;\xi }}_{k+1}\leq \chi_{\gamma }^{2}\left(m\right)$$

### 3.3. Adjustment of Measurement Noise Covariance Matrix

It is generally believed that we should pay attention to the position of recent measurements in the current filtering in order to prevent filtering divergence; hence, we need to pay special attention to the innovation sequence [18,19,20,21].

In the filtering process of the improved SHAEKF, we used (Equation 21) to judge the filtering state. If (Equation 21) was not tenable, this indicated that there was measurement abnormality and it was necessary to re-estimate the measurement noise parameters. If (Equation 21) held, there was no need to re estimate the measured noise parameters. In the conventional Sage–Husa adaptive filtering algorithm, the measurement noise parameters were adaptively estimated by the equal-weighted time average:(22)$${\hat{R}}_{k+1}^{lj}=\left(1-\frac{1}{k+1}\right){\hat{R}}_{k}^{lj}+\frac{1}{k+1}\left(s_{k+1}^{lj}{\left(s_{k+1}^{lj}\right)}^{T}-H_{k+1}^{j}P_{k+1|k}^{j}{\left(H_{k+1}^{j}\right)}^{T}\right)$$

There is a negative sign in the above formula. In order to prevent the measurement noise estimation from losing its positive definiteness, the measurement noise covariance matrix $R_{k+1}^{lj}$ is estimated based on the maximum a posteriori (MAP) estimation criterion:(23)$${\hat{R}}_{k+1}^{lj}=\frac{1}{k+1}\sum_{i=1}^{k+1}\left(Z_{i}^{lj}-H_{i}^{j}{\hat{X}}_{i|k+1}^{j}\right){\left(Z_{i}^{lj}-H_{i}^{j}{\hat{X}}_{i|k+1}^{j}\right)}^{T}.$$

Replace ${\hat{X}}_{i|k+1}^{j}$ with ${\hat{X}}_{i|i}^{j}$ to obtain the following approximation:(24)$$\begin{array}{c} \begin{array}{cc} Z_{i}^{lj}-H_{i}^{j}{\hat{X}}_{i|k+1}^{j} & \approx Z_{i}^{lj}-H_{i}^{j}{\hat{X}}_{i|i}^{j}\\ \; & =Z_{i}^{lj}-H_{i}^{j}\left({\hat{X}}_{i|i-1}^{j}+K_{i}^{j}s_{i}^{lj}\right)\\ \; & =\left(I-H_{i}^{j}K_{i}^{j}\right)s_{i}^{lj}. \end{array} \end{array}$$

If (Equation 24) is substituted into (Equation 23), we get the following equation:(25)$${\hat{R}}_{k+1}^{lj}=\frac{1}{k+1}\sum_{i=1}^{k+1}\left(I-H_{i}^{j}K_{i}^{j}\right)s_{i}^{lj}{\left(s_{i}^{lj}\right)}^{T}{\left(I-H_{i}^{j}K_{i}^{j}\right)}^{T}$$

It is easy to verify that the mean value of ${\hat{R}}_{k+1}^{lj}$ is as follows:(26)$$E\left({\hat{R}}_{k+1}^{lj}\right)=R_{k+1}^{lj}-\frac{1}{k+1}\sum_{i=1}^{k+1}H_{i}^{j}P_{i|i}^{j}{\left(H_{i}^{j}\right)}^{T}$$

The suboptimal MAP estimates of the measurement noise covariance matrix are obtained from (Equation 25) and (Equation 26):(27)$${\hat{R}}_{k+1}^{lj}=\frac{1}{k+1}\sum_{i=1}^{k+1}\left(\left(I-H_{i}^{j}K_{i}^{j}\right)s_{i}^{lj}{\left(s_{i}^{lj}\right)}^{T}{\left(I-H_{i}^{j}K_{i}^{j}\right)}^{T}+H_{i}^{j}P_{i|i}^{j}{\left(H_{i}^{j}\right)}^{T}\right)$$

Considering that $K_{k+1}^{j}$ and $P_{k+1}^{j}$ must be obtained after $R_{k+1}^{lj}$ and become stable during the filtering process, $K_{k}^{j}$ and $P_{k+1|k}^{j}$ can be used here to approximate $K_{k+1}^{j}$ and $P_{k+1}^{j}$ respectively.

From (Equation 27), the estimation of ${\hat{R}}_{k+1}^{lj}$ could be improved by using the exponential fading memory weighted average recurrence method in the following equation:(28)$$\begin{array}{c} \begin{array}{cc} {\hat{R}}_{k+1}^{lj} & =\left(1-\beta_{k+1}\right){\hat{R}}_{k}^{lj}+\beta_{k+1}(\left(I-H_{k+1}^{j}K_{k}^{j}\right)s_{k+1}^{lj}\\ \; & \times {\left(s_{k+1}^{lj}\right)}^{T}{\left(I-H_{k+1}^{j}K_{k}^{j}\right)}^{T}+H_{k+1}^{j}P_{k+1|k}^{j}{\left(H_{k+1}^{j}\right)}^{T}) \end{array} \end{array}$$
(29)$$\beta_{k+1}=\left\{\begin{array}{cc} 1 & k=0\\ \frac{\beta_{k}}{\beta_{k}+b} & k>1 \end{array}\right.$$
where, *b* is the fading memory factor, 0.95<b<0.99. It could be seen that when *b* was larger, ${\hat{R}}_{k}^{lj}$ accounted for a larger proportion of the estimated value of ${\hat{R}}_{k+1}^{lj}$. After the improvement, if the estimated value of the last filter gain was less than 1, the estimated value of ${\hat{R}}_{k+1}^{lj}$ could be guaranteed to be positive and definite, which suppressed the possibility of filter divergence.

In addition, considering that the state posterior estimation $P_{k+1}^{j}=\left(I-K_{k+1}^{j}H_{k+1}^{j}\right)P_{k+1|k}^{j}$, in general the EKF algorithm could only be established when the optimal filter gain was used. When the filter was not always stable or when a non-optimal filter gain was used, an un-simplified state posterior estimation error covariance equation was required:(30)$$P_{k+1}^{j}=\left(I-K_{k+1}^{j}H_{k+1}^{j}\right)P_{k+1|k}^{j}{\left(I-K_{k+1}^{j}H_{k+1}^{j}\right)}^{T}+K_{k+1}^{j}{\hat{R}}_{k+1}^{lj}{\left(K_{k+1}^{j}\right)}^{T}$$

Figure 2 describes the flow chart of the improved SHAEKF, and evaluates the measurement anomaly using the calculated the Chi-square test statistic based on innovation. The detection threshold was determined according to the confidence of the Chi-square test, and the Chi-square test statistics exceeding the threshold were regarded as measurement anomalies. When an observation anomaly occurred, the suboptimal MAP estimation weighted by the exponential decay memory was used to adjust the observation noise covariance matrix online. When no observation anomaly occurred, the measurement noise covariance at the previous time was maintained. It could then adjust the measurement noise covariance adaptively and ensure the stability of the filter.

## 4. Simulation and Result Analysis

### 4.1. Monte Carlo Simulation

In order to verify the feasibility of the improved SHAEKF algorithm, a leader AUV and a follower AUV were simulated in this study. Monte Carlo method was used in the simulation, and the simulation parameters were set as shown in Table 1. The initial error covariance was set to $Q_{0}^{j}=diag\left(\begin{array}{cc} 0.025^{2} & 0.5^{2} \end{array}\right)$, $R_{0}^{j}=diag\left(\begin{array}{cc} 10^{2} & 10^{2} \end{array}\right)$, $P_{0}^{j}=100\cdot I_{4}$, where diag (·) represents a diagonal matrix, and $I_{4}$ is the identity matrix of order 4.

We designed the AUVs to move along a straight line or a curve at a uniform constant speed, the actual trajectory and the DR calculation results of each AUV in a simulation test of the whole simulation are shown in Figure 3. As shown in Figure 4, we study the filtering effects of the EKF and the improved SHAEKF with or without measurement anomaly sequence under the above conditions. As shown in Figure 5a,b, in the steady state, we assumed that the observed values of the measurement model were continuously disturbed by a normally distributed random sequence with mean value of 0 and standard deviation of 10. Under the above simulation conditions, we studied the filtering effects of the EKF and the improved SHAEKF with and without a measurement anomaly sequence. As shown in Figure 5c,d, the normally distributed random anomaly sequence with a mean value of 0 and standard deviation of 150 was added to the observation values of the measurement models in the time period of 200–1200 s. Then, the EKF and the improved SHAEKF algorithms were used for the simulation by calculating the Chi-square test statistics based on innovation as the judgment index. Considering that the measurement model in this study involved relative distance, relative direction angle, and heading angle, the degree of freedom was taken to be m=3 and the confidence level was set at γ=0.01. The results of evaluating the measurement anomalies are shown in Figure 6. As shown in Figure 7 and Figure 8, the localization results of DR, EKF, and improved SHAEKF could be obtained respectively, including the root mean square and the standard deviation of the localization errors. The DR results of the follower AUV were omitted in Figure 7b,d,f,h and Figure 8b,d,f,h in order to more intuitively compare the localization errors of the EKF and the improved SHAEKF.

### 4.2. Comparison and Analysis of Simulation

Figure 3 shows that the localization errors of the follower AUV became larger and larger with the passage of time only based on DR, and the localization errors of the follower AUV were much larger than that of the leader AUV.

As can be seen from Figure 4, after obtaining the relative distance, relative direction angle, and high-precision DR localization data of the leader AUV, the follower AUV could improve its localization accuracy by using the EKF or the improved SHAEKF cooperative localization algorithm. It can be seen from Figure 4b,d, that when there were measurement anomalies, the filtering stability of the EKF was not as stable as that of the improved SHAEKF.

It can be seen from Figure 5 and Figure 6 that after adding the measurement anomaly sequence, the measurement model vector components of the follower AUV were disturbed and the measurement anomaly could be evaluated by calculating the Chi-square test statistics based on innovation.

In order to evaluate the performance of the improved algorithm more effectively, we introduced the average root mean square of localization error (ARMSE) and the average standard deviation of localization error (ASDE). ARMSE and ASDE can be defined as follows:(31)$$ARMSE=\frac{1}{n_{T}}\sum_{k=1}^{n_{T}}\left(\sqrt{\frac{\sum_{i=1}^{n_{MC}}{\left({\left(x_{k}-{\hat{x}}_{k}\right)}^{2}+{\left(y_{k}-{\hat{y}}_{k}\right)}^{2}\right)}_{i}}{n_{MC}}}\right)$$
(32)$$ASDE=\frac{1}{n_{T}}\sum_{k=1}^{n_{T}}\left(\sqrt{\frac{1}{n_{MC}-1}\cdot \sum_{i=1}^{n_{MC}}{\left(\sqrt{{\left({\left(x_{k}-{\hat{x}}_{k}\right)}^{2}+{\left(y_{k}-{\hat{y}}_{k}\right)}^{2}\right)}_{i}}-\frac{\sum_{i=1}^{n_{MC}}\sqrt{{\left({\left(x_{k}-{\hat{x}}_{k}\right)}^{2}+{\left(y_{k}-{\hat{y}}_{k}\right)}^{2}\right)}_{i}}}{n_{MC}}\right)}^{2}}\right)$$
where $n_{T}$ and $n_{MC}$ represent the sampling times and the Monte Carlo simulation times, respectively. ARMSE and ASDE respectively reflect the localization accuracy and the discreteness of the localization error dataset (i.e., the stability of the localization algorithm).

As can be seen from Figure 7, both the conventional EKF and improved SHAEKF work well if no measurement abnormality occurred. In this case, compared with the conventional EKF, the performance of the improved SHAEKF was not reduced, which meant that the contribution of these good measurements to the cooperative localization effect was not reduced, indicating the feasibility of the improved SHAEKF. It can be seen from Figure 8, when there were outliers in the measurement, the root mean square value and standard deviation of the localization errors obtained by EKF increased, and the root mean square value and standard deviation of the localization errors obtained by the improved SHAEKF were reduced to a certain extent. The improved SHAEKF could effectively resist the influence of outliers and had better performance than the EKF in localization accuracy, which indicated that the improved SHAEKF was more stable than the EKF. The above results could also be obtained by analyzing the data in Table 2 and Table 3.

## 5. Conclusions

The conventional EKF was only the best estimator when some preconditions were true, for example, when the Gaussian distribution measurement noise had a completely known mean and covariance. However, in the practical application of multi-AUV cooperative localization, the actual observation might be vulnerable to outliers. Therefore, a leader–follower multi-AUV adaptive cooperative localization filtering method based on innovation was proposed in this paper. Simulations showed that the improved SHAEKF could effectively resist the influence of measurement anomalies. Although the improved SHAEKF could ensure the filtering fault tolerance and localization accuracy of a leader AUV and a follower AUV in the case of abnormal measurement, we did not further study the influence of abnormal process noise on cooperative localization. In the future, we will further study the cooperative navigation and localization of multiple leader AUVs and multiple follower AUVs and expand from a two-dimensional plane to a three-dimensional space. All AUVs in the group were regarded as a unified system to improve the navigation and localization system model. AUVs exchanged their information, including measured and estimated position data, we then proposed an interactive multi-model fusion estimation method for future work to correctly estimate some parameters that could not be accurately obtained in the AUV dynamic model, including multi-observation information and multi-motion model information, so as to increase the robustness of collaborative navigation and localization.

## Figures and Tables

**Figure 1 sensors-22-05016-f001:**
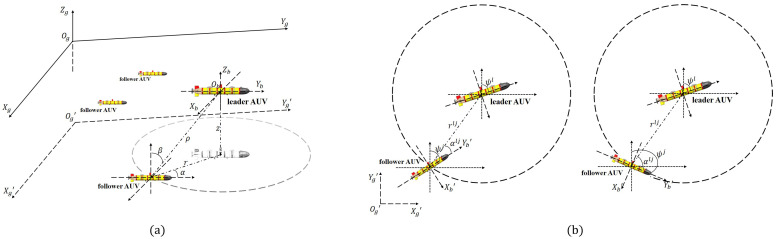
Schematic diagram of a multi-AUV cooperative localization system. (**a**) Description of the projection of the leader AUV in the $O_{g}-X_{g}Y_{g}Z_{g}$ coordinate system with respect to the ${O_{g}}^{\prime }-{X_{g}}^{\prime }{Y_{g}}^{\prime }$ horizontal plane in the ${O_{g}}^{\prime }-{X_{g}}^{\prime }{Y_{g}}^{\prime }{Z_{g}}^{\prime }$ coordinate system; (**b**) description of the relative position of the leader AUV and the follower AUV in the ${O_{g}}^{\prime }-{X_{g}}^{\prime }{Y_{g}}^{\prime }$ horizontal plane.

**Figure 2 sensors-22-05016-f002:**
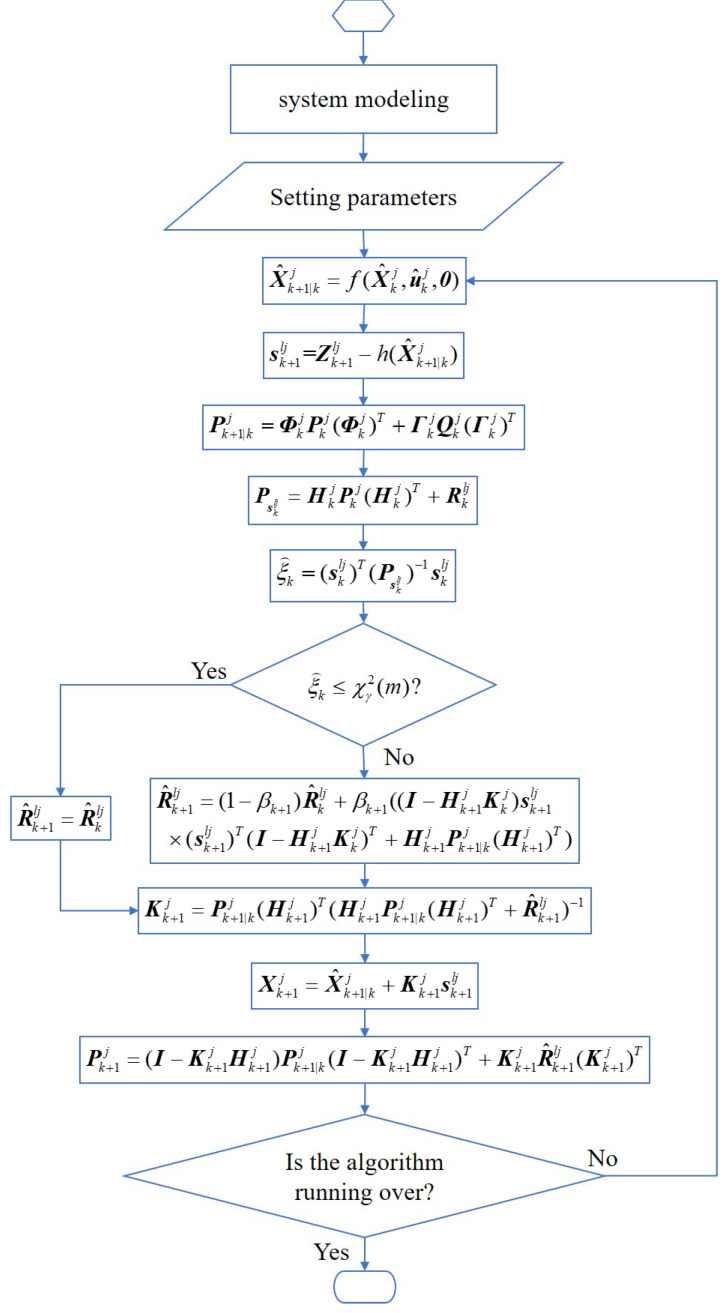
Flow diagram of the improved SHAEKF algorithm.

**Figure 3 sensors-22-05016-f003:**
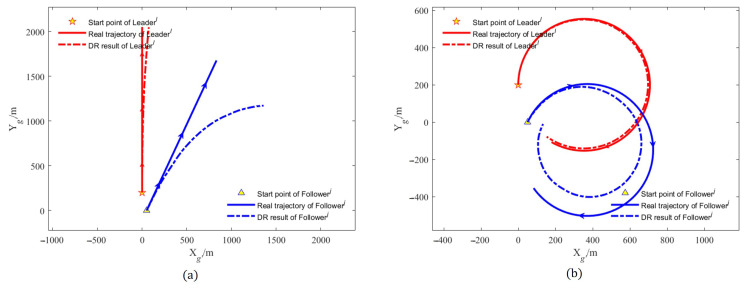
Trajectory diagram of ${Leader}^{l}$ and ${Follower}^{j}$ in a test. (**a**) AUVs moved along a straight line at a uniform constant speed; (**b**) AUVs moved along a curve at a uniform constant speed.

**Figure 4 sensors-22-05016-f004:**
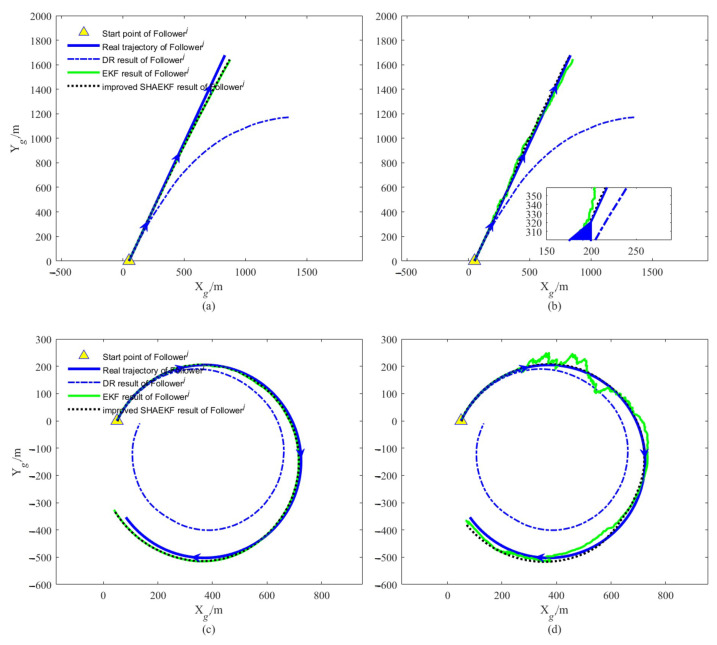
Trajectory diagram of ${Follower}^{j}$ in a test. (**a**,**c**) The localization results of each algorithm without measurement anomaly are described; (**b**,**d**) the localization results of each algorithm in the presence of measurement anomalies are described.

**Figure 5 sensors-22-05016-f005:**
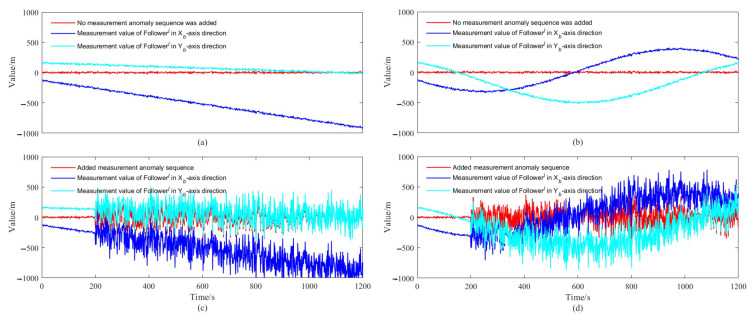
Measurement anomaly sequence and the components of the measurement model. (**a**,**c**) Description of the results of AUVs that moved along a straight line at a uniform constant speed; (**b**,**d**) description of the results of AUVs that moved along a curve at a uniform constant speed.

**Figure 6 sensors-22-05016-f006:**
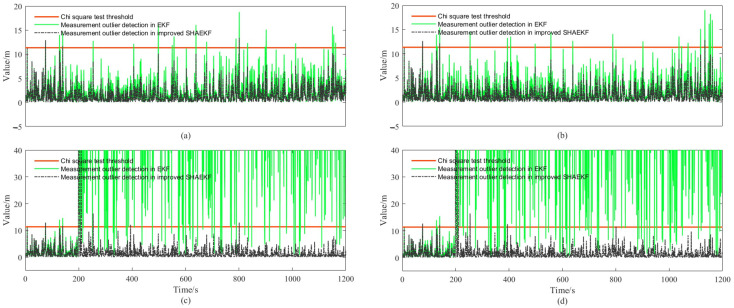
Detection of abnormal measurement. (**a**,**c**) Description of the results of AUVs that moved along a straight line at a uniform constant speed; (**b**,**d**) description of the results of AUVs that moved along a curve at a uniform constant speed.

**Figure 7 sensors-22-05016-f007:**
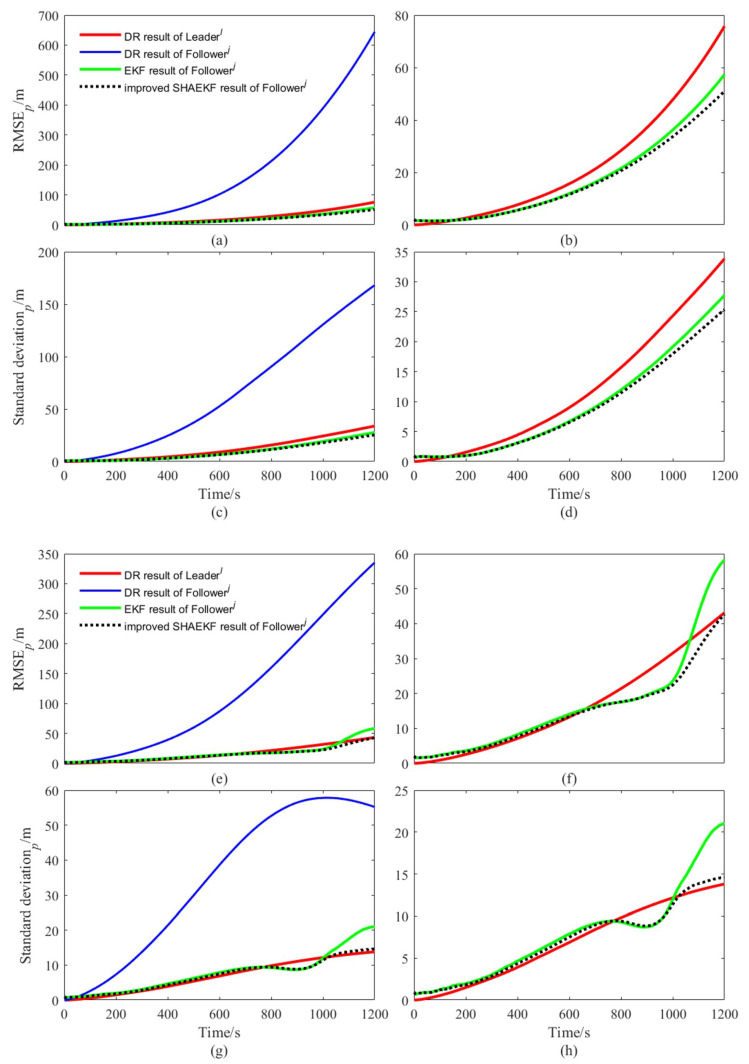
Comparison diagram of localization errors under a steady-state system. (**a**–**d**) Description of the results of AUVs that moved along a straight line at a uniform constant speed; (**e**–**h**) description of the results of AUVs that moved along a curve at a uniform constant speed; (**a**,**b**,**e**,**f**) description of the change in the root mean square of localization errors with time for each algorithm; (**c**,**d**,**g**,**h**) description of the variation of the standard deviation of localization errors with time for each algorithm.

**Figure 8 sensors-22-05016-f008:**
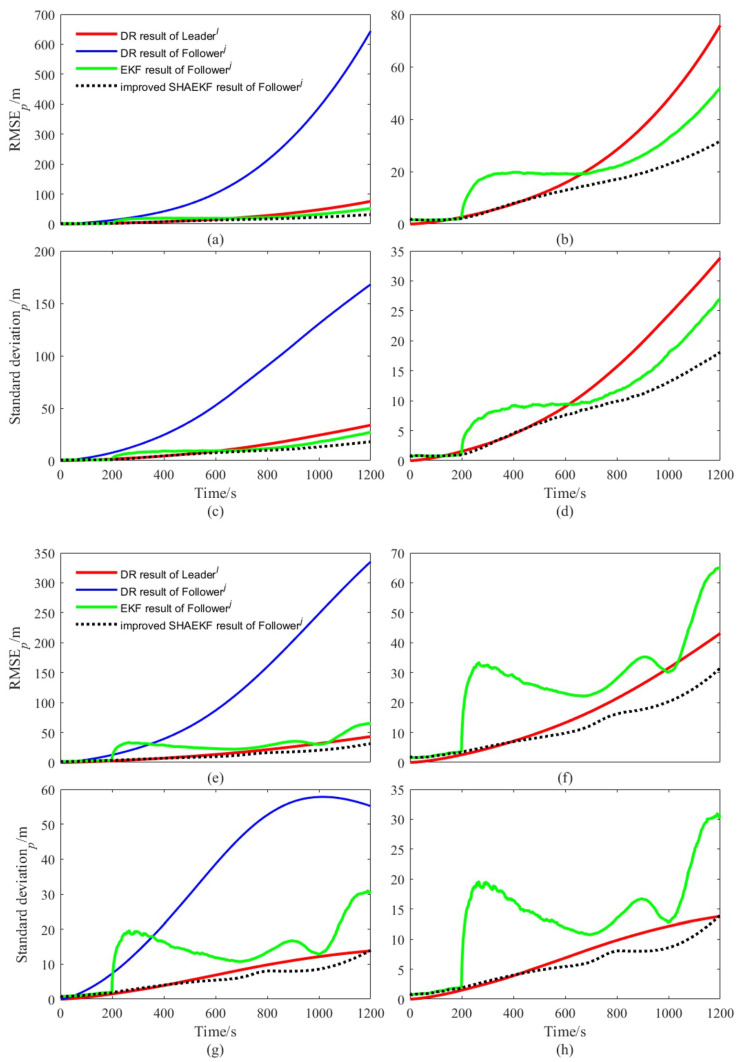
Comparison diagram of localization errors in case of measurement abnormality. (**a**–**d**) Description of the results of AUVs that moved along a straight line at a uniform constant speed; (**e**–**h**) description of the results of AUVs that moved along a curve at a uniform constant speed; (**a**,**b**,**e**,**f**) description of the change in the root mean square of localization errors with time for each algorithm; (**c**,**d**,**g**,**h**) description of the variation of the standard deviation of localization errors with time for each algorithm.

**Table 1 sensors-22-05016-t001:** Simulation parameter setting.

Simulation Parameters		Parameter Values
*Simulation time*, s		1200
*Sampling frequency*, Hz		1
*Monte Carlo simulation times*		1000
*Leader AUV*	Start point coordinates, m	(0, 200)
	Forward motion velocity, (m · s^−1^)	3× (1852/3600)
	Start point heading angle, rad	0×(π/180)
	Random error of forward motion velocity, (m · s^−1^)	±0.005
	Random error of heading angle, (rad · s^−1^)	±0.1
	Gyro bias, (rad · h^−1^)	0.03×(π/180)
*Follower AUV*	Start point coordinates, m	(50, 0)
	Forward motion velocity, (m · s^−1^)	3×(1852/3600)
	Start point heading angle, rad	25×(π/180)
	Random error of forward motion velocity, (m · s^−1^)	±0.025
	Random error of heading angle, (rad · s^−1^)	±0.5
	Gyro bias, (rad · h^−1^)	0.3×(π/180)

**Table 2 sensors-22-05016-t002:** Comparison of ARMSE of each algorithm.

Simulation Conditions in the Time Period 200–1200 s	ARMSE (m) of ${\mathrm{Follower}}^{j}$ When AUVs Moved Along a Straight Line			ARMSE (m) of ${\mathrm{Follower}}^{j}$ when AUVs Moved Along a Curve		
	*DR*	*EKF*	*Improved SHAEKF*	*DR*	*EKF*	*Improved SHAEKF*
Steady state	176.3030	17.4726	16.3846	118.1430	16.2329	14.3517
Abnormal measurement	176.3030	20.9517	13.0884	118.1430	27.1199	11.9600
Percentage change (%)	0	19.9	−20.1	0	67.1	−16.7

**Table 3 sensors-22-05016-t003:** Comparison of ASDE of each algorithm.

Simulation Conditions in the Time Period 200–1200 s	ASDE (m) of ${\mathrm{Follower}}^{j}$ when AUVs Moved Along a Straight Line			ASDE (m) of ${\mathrm{Follower}}^{j}$ When AUVs Moved Along a Curve		
	*DR*	*EKF*	*Improved SHAEKF*	*DR*	*EKF*	*Improved SHAEKF*
Steady state	64.5837	9.2438	8.7528	34.3510	7.7098	6.9763
Abnormal measurement	64.5837	10.6673	7.5491	34.3510	13.7229	5.7479
Percentage change (%)	0	15.4	−13.8	0	78.0	−17.6

## Data Availability

Not applicable.

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
