# Peer review of "An Adaptive Filtering Method for Cooperative Localization in Leader–Follower AUVs"

_sensors, 2022, doi:10.3390/s22135016_

Round 1

Reviewer 1 Report

This manuscript presents a filtering algorithm for multi-AUV cooperation localization in a master-slave pattern. A numerical simulation was also conducted to verify its accuracy and reliability. I think this study is interesting and the manuscript is basically well prepared, thus I would like to recommend its publication.

I have a few concerns as follows:

1.     The relationship between AUVs is termed as “master-slave” somewhere and “leader-follower” elsewhere. Why not keep using a consistent term?

2.     There are multiple leader AUVs and follower AUVs in the algorithm. Do they have a specified or optimal proportion in a AUV group? If the number of leader AUV increases, can the accuracy of localization improve? Will anything change when both leader and follower AUV number increase? The numerical simulation only modeled a case with a single leader and a single follower, which is considered insufficient for me. The multiple AUV case should be investigated, at least discussed in a discussion section.

3.     In table 2, I do not understand how those percentage numbers on the right hand side of the table were calculated. Could you better explain them?

Meanwhile, In the middle column, should “ASD reduction %” be “ASDE reduction % ?

Reviewer 2 Report

Dear Authors

Your paper addresses an interesting topic. It has the potential to become a good paper.

However, I suggest correcting a few deficiencies.

1. Captions of figures need to be extended to explain the figure clearly. It is OK to include the explanation in the text. For example, Figure 1 is complex, and the corresponding caption is only a half of one line long. There is also a strange small space between the caption and the text of line 85.

The same problem is present in Figure 2. There is poor caption description and no clear distinction between caption and text in line 114.

The same is true in Figure 3 and the rest of the figures

It would help to edit all figures and all captions for clarity

Figure units need to be edited to comply with standard rules

2. in line 72, you introduced an acronym ENU, which is not generally known.

3. Your algorithm description is boring for most readers. Please reduce it to mention only your improvements to SHAEKF. The readers are generally not interested in published and known facts.

They want to understand the results and what is an improvement compared to state of the art.

It would be better if everything were not based only on simulations, and more data would be based on actual measurements. It is difficult and expensive to run such an experiment in the real world, but how can you set test model parameters like signal-to-noise values or deterministic parameters like disturbances. I also believe that the Monte Carlo analysis of 50 repetitions is not enough to obtain an excellent statistical insight.

I propose making the results section more convincing by adding a simulation model with more realistic data.

I wish you good luck

Reviewer 3 Report

This paper presents a solid work on the cooperative localization problem. The method can largely reduce the localization errors compared to the state-of-the-art. The result is satisfactory and has no obvious flaws.

Round 2

Reviewer 2 Report

Dear Authors

Congratulations. The paper is now improved, and deserves to be published.